# Brain Fog and Quality of Life at Work in Non-Hospitalized Patients after COVID-19

**DOI:** 10.3390/ijerph191912816

**Published:** 2022-10-06

**Authors:** Zaneta Chatys-Bogacka, Iwona Mazurkiewicz, Joanna Slowik, Monika Bociaga-Jasik, Anna Dzieza-Grudnik, Agnieszka Slowik, Marcin Wnuk, Leszek Drabik

**Affiliations:** 1Department of Neurology, Jagiellonian University Medical College, 30-688 Krakow, Poland; 2Department of Neurology, University Hospital in Krakow, 30-688 Krakow, Poland; 3Department of Periodontology, Preventive Dentistry and Oral Medicine, Institute of Dentistry, Faculty of Medicine, Jagiellonian University Medical College, 31-155 Krakow, Poland; 4Department of Infectious and Tropical Diseases, Jagiellonian University Medical College, 30-688 Krakow, Poland; 5Department of Internal Medicine and Gerontology, Jagiellonian University Medical College, 30-688 Krakow, Poland; 6Department of Pharmacology, Jagiellonian University Medical College, 31-531 Krakow, Poland; 7John Paul II Hospital, 31-202 Krakow, Poland

**Keywords:** COVID-19, brain fog, life quality, employment, work, long COVID

## Abstract

*Background:* There is still a need for studies on the quality of life (QoL) at work among COVID-19 survivors. Therefore, we aimed to evaluate the association between the brain fog symptoms and the QoL at work in non-hospitalized patients with previous SARS-CoV-2 infection. *Methods:* Three hundred non-hospitalized patients (79.33% women; median age, 36 years; interquartile range, 30–48 years) were included in the final analysis. An anonymous neuropsychological questionnaire containing eight different questions on the presence of brain fog symptoms in four time intervals, i.e., pre-COVID-19 and 0–4, 4–12, and >12 weeks after infection, was retrospectively introduced to patients and staff of the University Hospital in Krakow. Additionally, a four-point Likert scale was used to evaluate QoL at work in four time periods. Included were participants aged ≥ 18 years in whom the diagnosis of COVID-19 was confirmed by the RT-PCR from nasopharyngeal swab and the first symptoms occurred no earlier than 3 months before the completion of the questionnaire. *Results:* Before SARS-CoV-2 infection, 28.00% (*n* = 84) of patients reported poor QoL at work. Within 4, 4–12, and >12 weeks after infection, a decrease in QoL was observed in 75.67% (*n* = 227), 65.00% (*n* = 195), and 53.66% (*n* = 161) of patients, respectively (*p* < 0.001). With increasing deterioration of the QoL at work, the number of brain fog symptoms increased, and patients with severe QoL impairment exhibited a median of five symptoms for <4, 4–12, and >12 weeks post-COVID-19. In the multivariable logistic regression model, predictors of the deterioration of the QoL at work depended on the time from COVID-19 onset; in the acute phase of the disease (<4 weeks), it was predicted by impairment in remembering information from the past (OR 1.88, 95%CI: 1.18–3.00, *p* = 0.008) and multitasking (OR 1.96, 95%CI: 1.48–2.58, *p* < 0.001). Furthermore, an impairment in the QoL at work 4–12 weeks and >12 weeks after COVID-19 was independently associated with age (OR 0.46, 95%CI: 0.25–0.85, *p* = 0.014 and OR 1.03, 95%CI: 1.01–1.05, *p* = 0.025, respectively), problems with multitasking (OR 2.05, 95%CI: 1.40–3.01, *p* < 0.001 and OR 1.75, 95%CI: 1.15–2.66, *p* = 0.009, respectively), answering questions in an understandable/unambiguous manner (OR 1.99, 95%CI: 1.27–3.14, *p* = 0.003 and OR 2.00, 95%CI: 1.47–2.36, *p* = 0.001, respectively), and, only for the >12 week interval, problems with remembering information from the past (OR 2.21, 95%CI: 1.24–3.92, *p* = 0.007). *Conclusions:* Certain brain fog symptoms, such as impaired memory or multitasking, are predictors of a poorer QoL at work not only during the acute phase of COVID-19 but also within more than 12 weeks after the onset of infection.

## 1. Introduction

The coronavirus disease 2019 (COVID-19) pandemic is still ongoing, with an increasing number of affected patients [1]. Coronavirus spreads easily through respiratory droplets and close contact [2]. Both cellular and humoral immunity play important roles in the protection against this infection [3], and vaccines have decreased viral transmission by inducing a robust immunological response [4]. However, among COVID-19 survivors, a substantial number of individuals still experience persistent neuropsychological symptoms [5].

Brain fog, which manifests with memory and concentration difficulties, is one of the most commonly reported sequelae after severe acute respiratory syndrome coronavirus 2 (SARS-CoV-2) infection [6]. A previous retrospective Dutch study showed that 8 weeks after hospitalization due to COVID-19, 33% and 27% of patients reported memory and concentration impairment, respectively, which correlated with the presence of depression and anxiety [7]. Post-acute complications of COVID-19 might also affect physical functioning and the ability to participate in social roles [8], with half of patients declaring deficits in at least one of the dimensions assessed with the Euro Quality of Life–Five Dimensions–Five Levels (EQ-5D-5L) scale, including usual activities [9].

In recent years, there has been a growing interest in research on patient-reported outcomes, including the quality of life (QoL) [10]. The authors of a recent meta-analysis of 151 studies from 32 countries underlined the need for further evaluation of post-COVID-19 syndrome, as at least one sequelae symptom was experienced by half of individuals up to 12 months after infection, with nearly one-fifth presenting with cognitive or memory impairments [11]. Research on the QoL in patients after SARS-CoV-2 infection produced divergent results and mostly concentrated on previously hospitalized individuals [12,13]. Some studies revealed reduced QoL a few weeks after hospital discharge [14], possibly due to decreased mobility [15]. On the other hand, 145 previously hospitalized Greek patients with COVID-19 reported good QoL compared with the general population; however, only a minority of participants experienced severe disease [16]. Moreover, as was shown in a recent large meta-analysis of more than 500 studies regarding long COVID, only 16% and 10% of them focused on the QoL and work aspects, respectively [17]. The authors of the cross-sectional survey of healthcare workers in South Africa speculated that symptoms after the acute phase of SARS-CoV-2 infection, including brain fog, could impact the QoL of the participants and their ability to perform work activities properly [18]. An anonymous Irish online questionnaire of 988 patients after COVID-19 showed that 38% of them reported severe limitations in their job [19]. Patients after COVID-19 not only exhibit decreased QoL but also report persistent cognitive problems affecting their ability to work [20]. It was recently shown that only 70% of individuals came back to work 4–8 weeks after hospitalization due to COVID-19, whereas nearly half of survivors showed a clinically significant decrease in the QoL [21]. A meta-analysis of six studies including 580 patients with previous SARS-CoV-2 infection revealed that one-quarter of them still did not return to work after a mean follow-up of 35 months [22]. On the other hand, in a longitudinal Chinese study evaluating 1192 COVID-19 survivors 2 years after hospitalization, it was shown that not only QoL improved in almost all domains, but also most patients (89%) returned to their original work [23]. However, in comparison to the general population, patients with previous SARS-CoV-2 infection experienced any persistent post-COVID-19 symptom twice as often [23]. Thus, a substantial number of patients after SARS-CoV-2 infection experience reduced QoL and concomitant difficulties in their daily activities, including participation in work [24]. However, it is unclear exactly which brain fog symptoms correlate with decreased QoL after COVID-19 and may potentially affect performance of duties at work.

Therefore, the aims of the current study were to assess the QoL at work among non-hospitalized COVID-19 survivors and its change within weeks after the onset of infection and to search for possible associations between the severity and number of brain fog symptoms and the QoL at work. We hypothesized that certain elements of brain fog could act as predictors of a poorer QoL at work in patients who experienced SARS-CoV-2 infection.

## 2. Materials and Methods

### 2.1. Participants and Data Collection

The study included patients aged 18 or older, with more than 3 months since the onset of SARS-CoV-2 infection, who were able to write and read. Data were collected between 22 April and 9 August 2021.

### 2.2. Assessment of the Brain Fog Symptoms and the Quality of Life

In order to prepare a short clinical questionnaire evaluating brain fog symptoms, we first searched the PubMed database for assessment tools and a spectrum of symptoms after COVID-19 [22,25,26]. Then, after an interview with 12 neurologists who experienced SARS-CoV-2 infection, and asking them open-ended questions on the presence of memory, concentration, sleep, and speech disturbances (Appendix A), we created an initial anonymous version of the Post-COVID Brain Fog (BF-COVID) questionnaire regarding these symptoms and their severity, administered in Polish. The BF-COVID questionnaire was then validated in a cohort of 70 individuals, including neurologists, other physicians, physiotherapists, neuropsychologists, and speech therapists. All members of the cohort completed the initial version of the questionnaire and gave their feedback. After further expert consensus, eight items regarding fatigue symptoms were excluded, leaving the final version of the BF-COVID questionnaire.

This retrospective questionnaire comprised 8 questions on the presence of difficulties in performing the following items: (1) writing, reading, and counting, (2) answering the questions in an understandable or unambiguous manner, (3) thoughts communicating during a conversation in a way that others can understand, (4) performing several independent tasks simultaneously, (5) recalling the new information, (6) remembering information from the past, (7) determining the current date and naming the days of the week, and (8) finding the right way in a familiar place. In addition, question number 9 evaluating the QoL at work (Quality of Life at Work Questionnaire) was added and comprised a 4-point Likert scale (0—normal, 1—mild, 2—moderate, and 3—severe impairment). This scale was previously used in other surveys regarding COVID-19 [27,28]. Two other sections were also added to the questionnaire and included (1) demographic data (age, sex) and (2) COVID-19-related data (date of SARS-CoV-2 infection diagnosis confirmed by the RT-PCR from a nasopharyngeal swab; the need for hospitalization).

The final version of the BF-COVID questionnaire was then introduced to patients who attended the post-COVID-19 ambulatory at the University Hospital in Krakow. These individuals were encouraged to fill out the paper version of the questionnaire anonymously. Moreover, an anonymous online version of the questionnaire was accessible through an invitation sent via mass e-mail to employees of the University Hospital in Krakow or through a link posted on Facebook. Participants completed the questionnaire once only retrospectively and declared the presence of symptoms in four time intervals, similarly to the guidelines of the National Institute for Health and Care Excellence (NICE) [29], that was as follows: pre-COVID-19, acute infection phase (i.e., 0–4 weeks since the COVID-19 onset), the post-acute phase (i.e., 4–12 weeks after infection), and the chronic phase (i.e., >12 weeks after infection). With reference to every time period, patients also reported their QoL at work.

We received a total of 660 BF-COVID questionnaires. From this number, excluded were those with incomplete data (*n* = 57) and those filled out by patients with a history of hospitalization due to COVID-19 (*n* = 303).

### 2.3. Psychometric Analysis

We performed exploratory factor analysis and reliability testing [30]. In the first step, we used the correlation matrix to exclude variables with an intercorrelation coefficient ≥ 0.8. Then, we evaluated the adequacy of the sampling with Bartlett’s sphericity test and the Kaiser–Meyer–Olkin (KMO) measure. As Bartlett’s *p*-value was <0.05 and KMO was >0.5, in the next step, we performed an exploratory factor analysis (EFA) with orthogonal rotation to assess the structure of the domains [31]. The number of items was based on the analysis of the scree plot and the Kaiser criterion (eigenvalue > 1) [32], and the cut-off value for factor loadings in the EFA was 0.6. Decisively, we identified three domains. We measured the internal consistency of the BF-COVID and the Quality of Life at Work Questionnaire with Cronbach’s alpha, and values ≥ 0.70 were considered acceptable [33].

### 2.4. Statistics and Bioethics

STATISTICA version 13.0 (Statsoft Inc., Tulsa, OK, USA) was used for the analysis. For more details, see Supplemental Methods—Statistics [34].

We performed the current study according to the Declaration of Helsinki and as part of the CRACoV-HHS project (CRAcow in CoVid pandemics—Home, Hospital and Staff), which received the Jagiellonian University Bioethics Committee approval [35]. As BF-COVID questionnaires were anonymously filled out by patients, data collection in the current study did not require additional approval from the Bioethics Committee after consulting a legal opinion.

## 3. Results

### 3.1. Psychometric Properties of the BF-COVID Questionnaire

The final version of the questionnaire was distributed to 300 participants to test psychometric properties. A bivariate correlation score for all items was acceptable (<0.8). The dates were suitable for EFA, as the KMO value was 0.796, and the Bartlett sphericity test remained significant (Chi^2^ = 943; df = 36, *p* value <0.001) for pre-COVID-19. We extracted three dimensions based on the scree plot and Kaiser’s criterion, which explained 57.14% of the total variance. The three dimensions were ‘communication and orientation’, ‘multitasking and quality of life at work’, and ‘memory’ (Appendix A). Finally, the corrected Cronbach alpha values were 0.740, 0.833, 0.829, and 0.833 for pre-COVID-19, 0–4 weeks, 4–12 weeks, and >12 weeks, respectively, indicating acceptable internal consistency of the BF-COVID questionnaire at four different time points.

### 3.2. The Quality of Life at Work before and after COVID-19

A total of 300 non-hospitalized patients, including 79.33% women, and a median age of 36 (IQR, 30–48) years, were enrolled in this study. The youngest and the oldest participants were 20 and 73 years old, respectively.

Before COVID-19, 28.00% (*n* = 84) of patients reported poor QoL at work, including 6.33% (*n* = 19) to a moderate or severe degree (Table 1).

The proportion of patients who experienced a change in QoL at work after SARS-CoV-2 infection compared to the pre-COVID-19 period is represented in Figure 1.

Within 4, 4–12, and >12 weeks after COVID-19, a decrease in the QoL at work was observed in 75.67% (*n* = 227), 65.00% (*n* = 195), and 53.66% (*n* = 161) of patients, respectively (*p* < 0.001). A moderate or severe decrease in the QoL at work was found in 46.67% (*n* = 140), 30.33% (*n* = 91), and 20.96% (*n* = 61) of patients for the specified time intervals.

### 3.3. Association between the Quality of Life at Work and the Number of Brain Fog Symptoms

Before COVID-19, patients with mild impairment of QoL at work had a higher number of symptoms of brain fog compared to those with normal QoL (a median of 1, interquartile range [0–3] vs. 0 [0–1], *p* < 0.001). However, for those with moderate or severe impairment of the QoL at work, the difference was statistically insignificant (Table 2, Appendix A).

After COVID-19, patients with any impairment in QoL at work (i.e., grade 1–3) had a higher number of brain fog symptoms compared to those with normal QoL in each of the time intervals evaluated. With increasing severity of deterioration of the QoL at work, the number of symptoms increased, and patients with severe impairment of the QoL were characterized by a median of five symptoms of brain fog after the onset of infection (Table 2, Appendix A).

### 3.4. Association between the Quality of Life at Work and Severity of Brain Fog Symptoms

Patients with deterioration of QoL at work had a higher prevalence of nearly all brain fog symptoms during all time periods after COVID-19 compared to those without changes in the QoL (Table 3).

We observed the strongest positive correlation (Spearman correlation coefficient of 0.4–0.6) between the QoL at work and problems with multitasking (question 4), answering the questions in an understandable/unambiguous manner (question 2), communication of thoughts (question 3), and recalling new information for each interval evaluated after COVID-19 (question 5, Figure 2).

This association was more pronounced with a longer duration of the disease and was relatively weak before the disease.

In the multivariable logistic regression model, the predictors of the deterioration of the QoL at work depended on the time from COVID-19 onset; in the acute phase of the disease (<4 weeks), it was predicted by impairment in remembering information from the past, multitasking, or recalling new information (models A and B, Table 4).

Then, an impairment in the QoL at work 4–12 weeks and >12 weeks after COVID-19 was independently associated with age, problems with multitasking, answering questions in an understandable or unambiguous manner, and (only for interval >12 weeks) remembering information from the past (model A), or age (only for interval >12 weeks), problems with remembering information from the past, recalling new information, and answering questions in an understandable or unambiguous manner (model B).

Models that included up to three brain fog symptoms showed acceptable discrimination for deterioration of the QoL at work in the acute phase of the disease (area under the curve (AUC) of 0.662–0.694); however, a greater ability to discriminate impairment of the QoL at work was observed for intervals of 4–12 weeks and >12 weeks (AUC of 0.689–0.767, Table 4).

### 3.5. Impairment of the Quality of Life at Work before COVID-19

Patients with an altered QoL at work before COVID-19 had less frequent changes in symptoms (for at least one severity level) than those with normal QoL at work before the disease (<4 weeks: 52.39% vs. 67.13%, p = 0.018; 4–12 weeks: 32.14% vs. 56.48%, *p* < 0.001; and >12 weeks: 25.37% vs. 46.86%, *p* < 0.001).

An impairment in the QoL at work before COVID-19 was associated with a lower risk of further worsening of QoL after SARS-CoV-2 infection (<4 weeks, odds ratio (OR) 0.54, 95% confidence interval (CI) 0.32–0.90; 4–12 weeks, OR 0.36, 95%CI 0.22–0.62; and >12 weeks OR 0.29, 95%CI 0.16–0.52).

## 4. Discussion

Our study is among the first to show that certain elements of brain fog, such as impaired memory and multitasking, are predictors of a poorer QoL at work not only during the acute phase of COVID-19 but also more than 12 weeks after disease onset. In addition, difficulties with answering questions in an understandable or unambiguous manner were independently associated with impairment of QoL 4 weeks after the onset of COVID-19. Therefore, we were able to confirm our working hypothesis. Until now, it was known that non-hospitalized patients who tested positive for SARS-CoV-2 showed worse attention and working memory a few months after infection compared to the demographically matched population [36]. Similar to our study, these residual post-COVID-19 deficits negatively affected the QoL [37] and, additionally, resulted in a greater risk of missing days at work [36]. Worse QoL after the acute phase of SARS-CoV-2 infection might also be accompanied by difficulties in the ability to think and engage in social activities, as was previously shown in a cohort of hospitalized individuals who required oxygen therapy on admission [38]. Although our study documented the correlation between brain fog symptoms and the QoL at work, previous research focused more on the rate of unemployment after COVID-19 [39]. It was shown that around one-third of patients were unable to return to work after a mean time of 110 days since SARS-CoV-2 infection and, interestingly, this also applied to the individuals with relatively mild disease [39,40]. A significant proportion of patients after COVID-19 could not return to work due to residual symptoms, including fatigue [41], and among those who were able to restart their job, 25% reported modified duties or reduced hours due to health status [42]. Therefore, it seems that post-COVID-19 symptoms, including brain fog, substantially affect both the QoL and the ability to work, as was previously shown also for central nervous system infections, such as pneumococcal meningitis or herpes simplex encephalitis [43]. The pathophysiological background of the correlation between the QoL and patient-reported difficulties in multitasking and memory that has been shown in our study remains to be determined. However, few studies using magnetic resonance imaging (MRI) and positron emission tomography (PET) in individuals after COVID-19 shed some light on this matter. Changes in the susceptibility-weighted imaging in brain MRI performed 2 to 3 months following the onset of infection were noted in the thalami of patients who also demonstrated decreased cognitive performance, especially in executive functions and visuospatial abilities [44]. Studies using PET in patients with post-COVID-19 syndrome showed that hypometabolism could be seen in the brainstem of individuals with long COVID, particularly in the locus coeruleus [45], that through noradrenaline neuromodulation had projections to the parts of the brain involved in cognition, such as the prefrontal cortex or hippocampus [46]. Other PET studies revealed that COVID-19 survivors with brain fog exhibited hypometabolism of the cingulate cortex that could explain deficits in episodic memory, executive functions, and attention and might be possibly related to delayed neuroinflammation [47]. Therefore, the results coming from our research could be potentially perceived as hypothesis-generating.

Although mild impairment in QoL was common before COVID-19, it became even more frequent and severe afterwards and only partially resolved 12 weeks after the onset of infection. The results of our study are in line with a recent meta-analysis of 12 studies with nearly 5000 patients indicating that post-COVID-19 syndrome was associated with poor QoL and persistent worse mental health [48]. In accordance with the results of our research, several previous studies also documented continuous recovery in the QoL within months after initial COVID-19 infection [49,50]. However, 33% and 40% of patients reported problems with word finding and concentration difficulties, respectively, within 12 months after the onset of SARS-CoV-2 infection [51]. Some studies followed a cross-sectional design similar to ours and also confirmed the high prevalence of residual brain fog symptoms several months after the onset of COVID-19 [52] and lower QoL among those with post-infection sequelae [53]. In a prospective Irish study of 155 patients, with demographic data similar to ours, i.e. mostly young women, of whom 55% did not require hospitalization, double assessment with the 12-Item Short Form Survey (SF-12) in a post-COVID-19 clinic, i.e., within 2–4 and 7–14 months after initial symptoms, revealed that the physical composite scores of the measured QoL significantly improved between two timepoints, but nevertheless remaining lower than the mean for the standardized healthy population [54]. Our study also revealed that mild, moderate, and severe impairment of QoL was associated with the severity of brain fog symptoms, as well as the number of symptoms. Furthermore, the correlation of neurological symptoms with the pre-COVID-19 impairment of the QoL was weak. Patients with normal QoL reported the highest deterioration in this issue, and paradoxically, an altered QoL before SARS-CoV-2 infection might have a protective effect. Previous studies showed that persistent post-COVID-19 symptoms could impact QoL, even among patients with initially mild illness [55]. Similar conclusions came from a Dutch study in which individuals with previously mild SARS-CoV-2 infection more frequently reported severe problems in comparison to discharged patients, especially related to physical functioning, QoL, and fatigue [56]. On the other hand, there are also studies showing that more serious post-COVID-19 symptoms, including pain and fatigue, were reported by patients with a more severe course of SARS-CoV-2 infection [24,57]. Therefore, even individuals with previously mild COVID-19 seem susceptible to decreased QoL with residual brain fog symptoms; however, whether the degree of these symptoms and the QoL could be associated with the severity of remote illness needs further research.

Our study revealed that older age independently affected QoL within more than 12 weeks since the onset of COVID-19, regardless of the used statistical model. Recent meta-analysis confirmed the role of age as a risk factor for post-COVID-19 residual symptoms [11]. The same conclusions came from a large Chinese study assessing patients 2 years after initial SARS-CoV-2 infection [23]. On the other hand, a prospective multicenter study of 90 patients assessed with the 36-Item Short Form Survey (SF-36) questionnaire 3 months after COVID-19 diagnosis revealed that one-third of individuals reported a reduction in QoL, which was further associated with younger age, longer hospitalization, impaired sleep, and anxiety [58]. Nevertheless, as was previously shown in a study of 91 critically ill COVID-19 patients who survived ARDS at the ICU in seven hospitals in Spain, a decrease in the QoL was seen in 67% of patients 6 months after hospitalization, and this was associated with advanced age apart from other factors [59]. Therefore, older age seems to increase the risk of development of brain fog symptoms after COVID-19 with accompanying lower QoL.

So far, the QoL in patients after COVID-19 has been assessed with different tools. In our study, due to its simplicity, we used a four-point Likert scale to measure QoL at work. This tool was also effectively used in other online surveys among participants who had COVID-19 [27,28]. Other authors evaluated QoL in individuals after COVID-19 with the Euro Quality of Life Visual Analogue Scale (EQ-VAS), where 0 and 100 meant the worst imaginable and the best health, respectively [14,40,60]. Other researchers used the SF-36 scale that consists of 36 statements and allows for the evaluation of the following eight elements: physical functioning, restrictions due to physical health, pain, general health sense, vitality, social functioning, emotional functioning, and mental health [49,61]. The SF-12 questionnaire is a validated and shortened version of the SF-36 scale, widely used in recent studies on the QoL in patients after SARS-CoV-2 infection [51,54]. More complicated scales evaluating different domains of the QoL, e.g., the Patient-Reported Outcome Measurement Information System (PROMIS) questionnaire, were also used and allowed for the assessment of the following issues: social role, pain, fatigue, physical function, and sleep [8,38]. Similarly, through the EQ-5D-5L questionnaire, patients after COVID-19 defined their health status from 1 to 5 in five different dimensions: mobility, self-care, usual activities, pain or discomfort, and anxiety or depression [9,15,21]. Current NICE guidelines do not recommend any specific scale for QoL assessment; however, according to them, it is more important to think about the impact of post-COVID-19 residual symptoms on the life of patients who survived SARS-CoV-2 infection [29].

Our research has some important limitations. First, the design of our study was retrospective and cross-sectional, relying on subjective responses of patients a few months after their initial infection. However, as shown in a previous study, self-reported neurological symptoms during and after COVID-19 could only be objectified in a limited way during neuropsychological testing [62]. Moreover, approximately half of our patients were public health workers familiar with possible neurological or psychiatric symptoms. Second, the QoL in patients after COVID-19 might also be related to symptoms other than brain fog, as was previously shown for insomnia [63], decreased lung diffusing capacity [64], frailty [65], and persistent olfactory and gustatory disturbances [66], among others. Third, no data regarding comorbidities that could additionally affect the presence of the post-COVID-19 symptoms [67] were gathered in our study.

## 5. Conclusions

In conclusion, certain brain fog symptoms, such as impaired memory and difficulties in multitasking, are predictors of poorer QoL at work 3 months after the onset of COVID-19. As previous studies have also reported decreased QoL and mental health problems among survivors of other epidemics, such as SARS-CoV-1 and Middle East Respiratory Syndrome [68], with a much higher prevalence of COVID-19 in comparison to the previous respiratory infections, even greater frequency of residual brain fog symptoms decreasing QoL is expected. Future studies, however, are needed to confirm the results of our study and also to concentrate on potential treatments [69,70,71].

## Figures and Tables

**Figure 1 ijerph-19-12816-f001:**
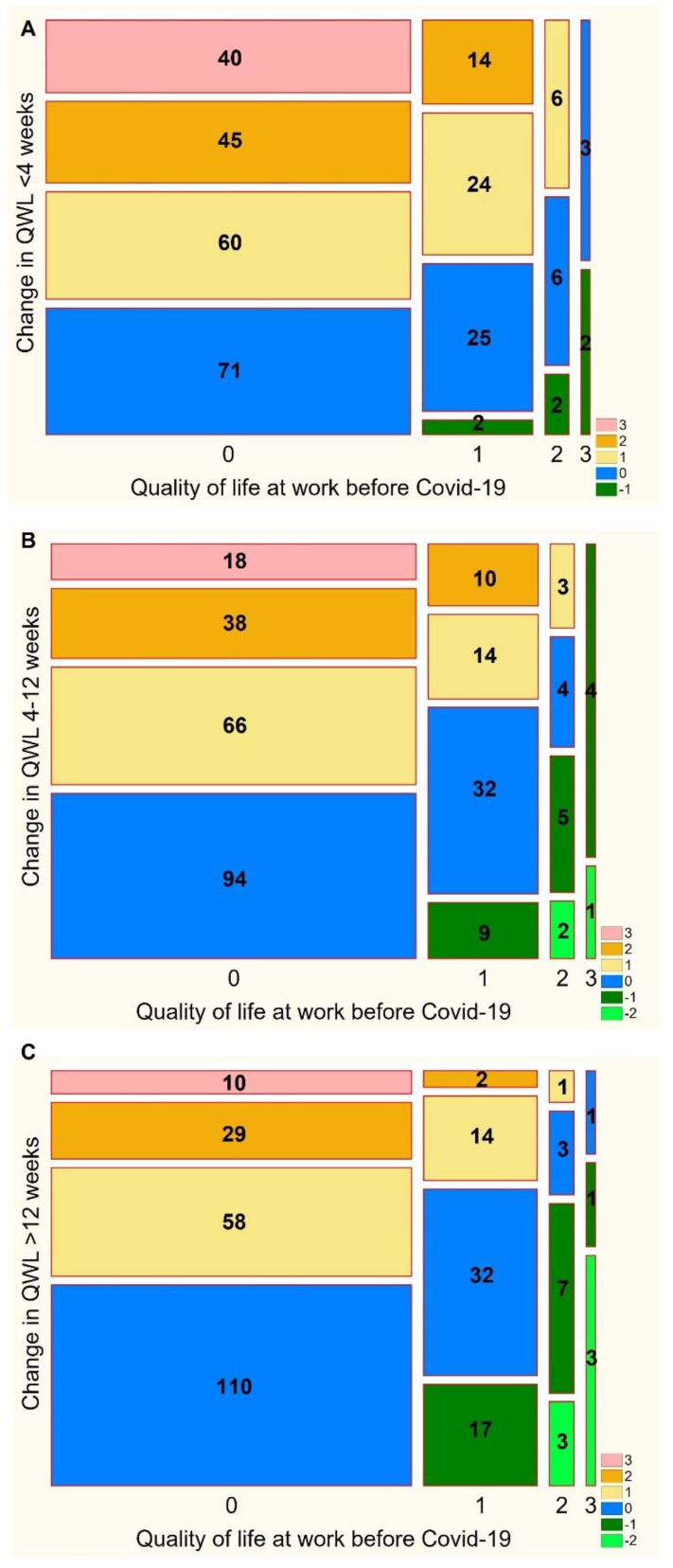
Mosaic plot. The proportion of patients with a change in the quality of life at work (QoL) <4 weeks (panel (**A**)), 4–12 weeks (panel (**B**)), and >12 weeks (panel (**C**)) after the onset of the SARS-CoV-2 infection compared to the pre-COVID-19 period. QoL was evaluated with a 4-point Likert scale, where 0—no symptoms, 1—mild, 2—moderate, and 3—severe symptoms. The numbers in bars correspond to the numbers of patients with changes by −2, −1, 0, 1, 2, or 3 points.

**Figure 2 ijerph-19-12816-f002:**
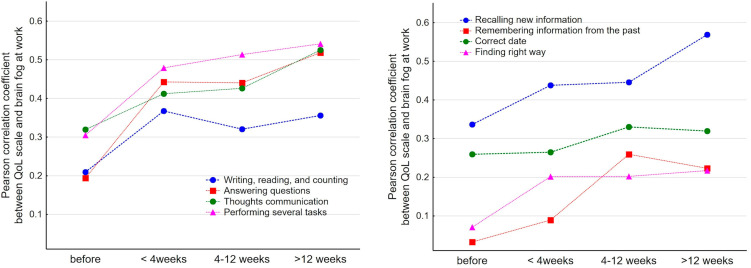
The correlation coefficients between brain fog symptoms and the quality of life at work during assessed time intervals after COVID-19.

**Table 1 ijerph-19-12816-t001:** Quality of life at work during four time intervals retrospectively assessed. Quality of life was evaluated with a 4-point Likert scale, where 0—no symptoms, 1—mild, 2—moderate, and 3—severe impairment.

Quality of Life at Work	Before COVID-19 (*n* = 300)	0–4 Weeks (*n* = 300)	4–8 Weeks (*n* = 300)	>12 Weeks (*n* = 291)	*p*-Value
Normal	216 (72.00)	73 (24.33)	105 (35.00)	130 (44.67)	<0.001
Mild impairment	65 (21.67)	87 (29.00)	104 (34.67)	100 (34.64)	
Moderate impairment	14 (4.67)	77 (25.67)	60 (20.00)	47 (16.15)	
Severe impairment	5 (1.67)	63 (21.00)	31 (10.33)	14 (4.81)	

**Table 2 ijerph-19-12816-t002:** Association between the number of symptoms and the quality of life at work in four time intervals: before COVID-19, in acute (<4 weeks), subacute (4–12 weeks), and chronic phases (>12 weeks). QoL was evaluated with a 4-point Likert scale, where 0 denoted no symptoms, 1—mild, 2—moderate, and 3—severe symptoms. Values are presented as median (interquartile range). Kruskal–Wallis test *p*-value was <0.05 for all four group comparisons. The post hoc Dunn test *p*-values < 0.05 are presented as * vs. no symptoms, # vs. mild symptoms.

Quality of Life at Work	Number of Symptoms of Brain Fog (median, IQR)			
	Before COVID-19	0–4 Weeks	4–12 Weeks	>12 Weeks
Normal	0 (0–1)	1 (0–3)	1 (0–2)	0 (0–1)
Mild impairment	1 (0–3) *	3 (1–4) *	3 (1–5) *	2 (1–4) *
Moderate impairment	1 (0–3)	4 (2–6) *#	4 (2–6) *	5 (2–6) *#
Severe impairment	0 (0–1)	5 (3–6) *#	5 (2–6) *#	5 (4–6) *#

**Table 3 ijerph-19-12816-t003:** Quality of life (QoL) at work and elements of brain fog during four assessed time intervals. The deterioration in QoL was defined as any increase (for ≥1 level on a 4-point Likert scale) compared to the values before COVID-19. For consecutive responses to the neurological symptoms questionnaire, 0—no symptoms, 1—mild impairment, 2—moderate impairment, and 3—severe impairment.

	Before COVID-19	Deterioration in QoL								
		0–4 Weeks			4–12 Weeks			>12 Weeks		
		No	Yes	*p*-Value	No	Yes	*p*-Value	No	Yes	*p*-Value
1. Writing, reading, and counting, *n* (%)										
0	294 (97.02)	94 (84.68)	124 (65.61)	<0.001	132 (87.42)	112 (75.17)	0.013	164 (92.66)	85 (7.56)	<0.001
1	5 (1.65)	10 (9.01)	42 (22.22)		15 (9.93)	20 (13.42)		11 (6.21)	21 (18.42)	
2	3 (0.99)	4 (3.60)	21 (11.11)		3 (1.99)	15 (10.07)		1 (0.56)	6 (5.26)	
3	0 (0.00)	3 (2.70)	2 (1.06)		1 (0.66)	2 (1.34)		1 (0.56)	2 (1.75)	
2. Answering the questions in an understandable or unambiguous manner, *n* (%)										
0	286 (94.38)	81 (72.97)	95 (50.26)	0.001	116 (76.82)	73 (48.99)	<0.001	152 (85.88)	61 (53.51)	<0.001
1	15 (4.95)	20 (18.02)	54 (28.57)		32 (21.19)	48 (32.21)		21 (11.86)	35 (30.70)	
2	1 (0.33)	9 (8.11)	36 (19.05)		2 (1.32)	24 (16.11)		4 (2.26)	16 (14.04)	
3	1 (0.33)	1 (0.90)	2 (2.12)		1 (0.66)	4 (2.68)		0 (0.00)	2 (1.75)	
3. Thoughts communicating during a conversation in a way that others can understand, *n* (%)										
0	263 (86.80)	65 (58.56)	79 (41.80)	0.009	105 (69.54)	63 (42.28)	<0.001	145 (81.92)	54 (47.37)	<0.001
1	37 (12.21)	32 (28.83)	57 (30.16)		41 (27.15)	55 (36.91)		27 (15.25)	41 (35.96)	
2	2 (0.66)	12 (10.81)	43 (22.75)		3 (1.99)	24 (16.11)		3 (1.69)	14 (12.28)	
3	1 (0.33)	2 (1.80)	10 (5.29)		2 (1.32)	7 (4.70)		2 (1.13)	5 (4.39)	
4. Performing several independent tasks simultaneously, *n* (%)										
0	257 (84.82)	71 (63.96)	62 (30.80)	<0.001	99 (65.56)	51 (34.23)	<0.001	137 (77.40)	49 (42.98)	<0.001
1	42 (13.86)	20 (18.02)	58 (30.69)		43 (28.48)	55 (36.91)		32 (18.08)	38 (33.33)	
2	3 (0.99)	18 (16.22)	43 (22.75)		7 (4.64)	34 (22.82)		6 (3.39)	22 (19.30)	
3	1 (0.33)	2 (1.80)	26 (13.76)		2 (1.32)	9 (6.04)		2 (1.13)	5 (4.39)	
5. Recalling new information, *n* (%)										
0	241 (79.54)	56 (50.45)	55 (29.10)	<0.001	88 (58.28)	44 (29.53)	<0.001	130 (73.45)	36 (31.58)	<0.001
1	50 (16.50)	31 (27.93)	50 (26.46)		48 (31.79)	59 (39.60)		34 (19.21)	46 (40.35)	
2	12 (3.96)	18 (16.22)	55 (29.10)		10 (6.62)	32 (21.48)		7 (3.95)	20 (17.54)	
3	0 (0.00)	6 (5.41)	29 (15.34)		5 (3.31)	14 (9.40)		7 (3.95)	12 (10.53)	
6. Remembering information from the past; for example, recognizing people or remembering events, *n* (%)										
0	266 (87.79)	89 (80.18)	125 (66.14)	0.025	126 (83.44)	105 (70.47)	0.003	157 (88.70)	81 (70.05)	<0.001
1	37 (12.21)	20 (18.02)	47 (24.87)		24 (15.89)	30 (20.13)		19 (10.73)	26 (22.81)	
2	0 (0.00)	2 (1.80)	14 (7.41)		1 (0.66)	12 (8.05)		0 (0.00)	6 (5.26)	
3	0 (0.00)	0 (0.00)	3 (1.59)		0 (0.00)	2 (1.34)		1 (0.56)	1 (0.88)	
7. Determining the current date and naming the days of the week, *n* (%)										
0	282 (93.07)	92 (82.88)	149 (78.84)	0.734	133 (88.08)	118 (79.19)	0.060	164 (92.66)	92 (80.70)	0.016
1	19 (6.27)	13 (11.71)	25 (13.23)		14 (9.27)	21 (14.09)		10 (5.65)	18 (15.79)	
2	1 (0.33)	4 (3.60)	12 (6.35)		4 (2.65)	5 (3.36)		3 (1.69)	3 (2.63)	
3	1 (0.33)	2 (1.80)	3 (1.59)		0 (0.00)	5 (3.36)		0 (0.00)	1 (0.88)	
8. Finding the right way in a familiar place, *n* (%)										
0	295 (97.36)	103 (92.79)	160 (84.66)	0.008	142 (94.04)	127 (85.23)	0.014	170 (96.05)	100 (87.72)	0.012
1	5 (1.65)	2 (1.80)	24 (12.70)		6 (3.97)	19 (12.75)		4 92.26)	13 (11.40)	
2	3 (0.99)	4 (3.60)	4 (2.12)		3 (1.99)	1 (0.67)		2 (1.13)	1 (0.88)	
3	0 (0.00)	2 (1.80)	1 (0.53)		0 (0.00)	2 (1.34)		1 (0.56)	0 (0.00)	

**Table 4 ijerph-19-12816-t004:** Predictors of the deterioration of quality of life (QoL) at work in acute (< 4 weeks), subacute (4–12 weeks), and chronic (>12 weeks) phases of COVID-19. The deterioration in QoL was defined as an increase for ≥1 level in a Likert scale compared to the pre-COVID-19 value.

	Univariable Analysis		Multivariable Analysis Model A		Multivariable Analysis Model B	
	OR (95% CI)	*p*-Value	OR (95% CI)	*p*-Value	OR (95% CI)	*p*-Value
<4 weeks after COVID-19						
Age (per year)	1.00 (0.98–1.02)	0.921	-	-	-	-
Female sex	1.30 (0.74–2.30)	0.367	-	-	-	-
Writing, reading, and counting (pp)	1.70 (1.15–2.50)	0.008	-	-	-	-
Answering the questions (pp)	1.89 (1.34–2.66)	<0.001	-	-	-	-
Thoughts communication (pp)	1.65 (1.23–2.21)	<0.001	x	x	x	x
Performing several tasks (pp)	2.00 (1.52 –2.64)	<0.001	1.96 (1.48–2.58)	<0.001	x	x
Recalling new information (pp)	1.73 (1.35–2.22)	<0.001	x	x	1.65 (1.28–2.13)	<0.001
Remembering information from the past (pp)	1.96 (1.25–3.07)	0.003	1.88 (1.18–3.00)	0.008	1.75 (1.10–2.80)	0.019
AIC			365.20		375.27	
AUC (95% CI)			0.694 (0.632–0.756)		0.662 (0.600–0.724)	
The Hosmer–Lemeshow test *p*-value			0.586		0.791	
4–12 weeks after COVID-19						
Age (per year)	1.02 (1.00–1.04)	0.041	0.46 (0.25–0.85)	0.014	-	-
Female sex	0.71 (0.40–1.25)	0.231	-	-	-	-
Writing, reading, and counting (pp)	1.88 (1.22–2.86)	0.003	-	-	-	-
Answering the questions (pp)	2.83 (1.92–4.20)	<0.001	1.99 (1.27–3.14)	0.003	2.63 (1.78–3.87)	<0.001
Thoughts communication (pp)	2.49 (1.75–3.55)	<0.001	x	x	x	x
Performing several tasks (pp)	2.63 (1.89–3.66)	<0.001	2.05 (1.40–3.01)	<0.001	-	-
Recalling new information (pp)	2.15 (1.61–2.89)	<0.001	x	x	1.41 (0.94–2.12)	0.097
Remembering information from the past (pp)	2.15 (1.37–3.38)	<0.001	-	-	1.87 (1.65–3.00)	0.010
Current date (pp)	1.69 (1.09–2.63)	0.020	-	-	-	-
Finding the right way (pp)	1.87 (1.01–3.450)	0.047	-	-	-	-
AIC			370.38		380.88	
AUC (95% CI)			0.727 (0.669–0.785)		0.689 (0.627–0.751)	
The Hosmer–Lemeshow test *p*-value			0.243		0.963	
>12 weeks after COVID-19						
Age (per year)	1.03 (1.01–1.06)	0.026	1.03 (1.01–1.05)	0.025	1.03 (1.01–1.06)	0.016
Female sex	1.07 (0.59–1.93)	0.828	-	-	-	-
Writing, reading, and counting (pp)	2.86 (1.63–5.03)	<0.001	-	-	-	-
Answering the questions (pp)	3.65 (2.32–5.74)	<0.001	2.00 (1.47–2.36)	0.001	2.42 (1.39–4.20)	0.002
Thoughts communication (pp)	3.12 (2.08–4.70)	<0.001	x	x	x	x
Performing several tasks (pp)	2.85 (1.98–4.09)	<0.001	1.75 (1.15–2.66)	0.009	-	-
Recalling new information (pp)	2.64 (1.92–3.62)	<0.001	x	x	1.70 (1.15–2.43)	0.008
Remembering information from the past (pp)	2.66 (1.57–4.52)	<0.001	2.21 (1.24–3.92)	0.007	2.01 (1.12–3.64)	0.021
Current date (pp)	2.20 (1.23–3.94)	0.008	-	-	-	-
AIC			334.86		334.58	
AUC (95% CI)			0.764 (0.707–0.822)		0.767 (0.709–0.825)	
The Hosmer–Lemeshow test *p*-value			0.944		0.288	

Model A included age, sex, and problems with (a) writing, reading, and counting, (b) answering questions in an understandable manner, (c) performing several independent tasks, (d) remembering information from the past, (e) determining the current date and day of the week (for models 4–12 and >12 weeks), and (f) finding the right way (model 4–12 weeks only). Model B included all variables from model A except for (c) problems with performing several independent tasks, which were replaced with problems with recalling new information. Abbreviations: AIC—Akaike information criterion, AUC—the area under the curve, CI—confidence interval, OR—odds ratio, pp—per point, x—a variable excluded from the multivariable model due to collinearity.

## Data Availability

Data confirming the results of the current study are available upon reasonable request from the corresponding author.

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
