# Peer review of "Brain Fog and Quality of Life at Work in Non-Hospitalized Patients after COVID-19"

_ijerph, 2022, doi:10.3390/ijerph191912816_

Round 1
Reviewer 1 Report
1. Regarding the explanation of the BF-COVID questionnaire, it is very confusing because it is dispersed and explained in 2.1. Please describe them all together in one section.
2. Is there any reason why you dare to say that the BF-COVID questionnaire was checked by 70 people, including health care professionals, for validation?
3. You mention that the final version of the BF-COVID questionnaire was answered by patients, university staff, etc. via social networking sites, what is the intent of this? Also, if it was conducted, is it not necessary to clearly state the results and explain how they compared to the results of the initial validation?
4. As for the order of the method descriptions, it would be easier to read if the description of the patient and data collection were first. Also, regarding the age range, you mention 18 and more years, but what was the oldest age? If you have information, it would be better to specify.
5. It is stated that 660 questionnaires were answered and 300 were used for the analysis. It only states in the data collection that those with incomplete answers or a history of hospitalization were excluded, but it would be more polite to include the relevant information for each.
Also, does it mean that the number of missing is almost half of the total number (n=660)? I don't know what the number of incomplete responses is, so I can't say, but simply looking at it, the number of missing responses seems to be quite large. Did you not consider using the multiple imputation method in your analysis?
6. Line 178, which questionnaire was used to ask about work-related matters?
7. It would be easier to understand the content if the tables and figures could be placed in the order of the results and near the text.
Author Response
The response to the Reviewer 1 is attached below.

Reviewer 2 Report
The manuscript is an original scientific paper that addresses brain fog symptoms in relation to quality of life within the sample of covid-19 (non-hospitalized) patients. While brain fog symptoms are one of the commonly reported outcomes of the covid-19 and play an important role within work environment, it is important to obtain new knowledge on the topic. Moreover, it is sensible to investigate in what relation it is with quality of life at work as it plays an important role. The authors have made a great work by preparing the manuscript, however, there are some inconsistencies. Some dilemmas and suggestions for improvement are listed below.
GENERAL COMMENTS:
· The manuscript needs to be proof-read.
INTRODUCTION:
· General comment: The introduction needs to be expanded. I suggest adding more ground information on brain fog and quality of life at work concepts. Introduce concepts more into details and provide literature review in a more comprehensive way. Emphasize results of the current studies that are crucial for you; namely serve as the grounds for carrying out present research.
· Derive the aim of the study more into details. Derive the aim more into details. Be more specific.
MATERIALS AND METHODS:
· Lines 104-105: It is not completely clear when the whole questionnaire was administered. As well when BF-COVID was administered to patients. I believe it was after its validation, which you mention? Please describe creation of the metric tool and data collection in a more comprehensive and clear manner.
· Line 109: Qql - How did u measure quality of life at work? Which tool did you use? Is this tool already validated in Polish language?
· Lines 112-113: I suggest dividing this to three sections. Namely 1) BF-COVID, which is, if I understand right, a stand-alone metric tool, then 2) a section with demographic data, such as age, sex, and any other similar data, and 3) a section with covid-19 related data, such as the date of its onset.
· Lines 114-116: Have you tested the questionnaire in the patients cohort? Did the patients understand all the statements? Did you test for the metric characteristics of the questionnaire?
· Line 126: Only BF-COVID questionnaire or demographic and covid-19 related data as well? Be more specific.
· Line 138: The majority of information within this subchapter is not necessary and can be limited to a minimum and mentioned briefly within the results section.
· Line 173-174: Data statement should be placed in the end of the manuscript, according to the instructions for authors.
RESULTS:
· General comment: present results in a more comprehensive manner.
· General comment: insert tables and figures after paragraphs in which you mention them.
· Figure 2 – it can be omitted. Results presented in a table would be more informative.
· Line 244: Table 1 – you report on normal, mild, moderate and severe impairment – it is not clear enough how did you measure and analyzed this – add this information in the methods section and describe it more into details.
DISCUSSION:
· General comment: Write discussion in a more structured and comprehensive way.
· Lines 268-314 and 315-345 and 346-389: Compare your results with other studies in a more comprehensive way. Elaborate more on your findings and less on other previous studies, support your findings with additional explanation on why the results might be like they are etc.
· Lines 390- 408: Elaborate on the metric tools used in a more comprehensive way. Also mind adding information on how you measured quality of life in the methods section.
Author Response

(The authors gave the same response as above.)

Reviewer 3 Report
Abstract, line 19: “after COVID-19”. This should be changed because the pandemic is not over and it is not correct to refer also that the investigation is scarce because there is already evidence in this field, but, on the other hand there is no ‘much’ time to have documented research concerning the period of crisis ongoign;
Abstract: “Three hundred non-hospitalized patients (79.33% women, 28 median age 36 [interquartile range, 30-48] years) were included in the final analysis” – this should be placed above in the abstract because participants information should be first in the method section.
Abstract: please reduce the results described because the abstract should be only the clear indicator of summarized data.
Introduction: clarify that the pandemic is still ongoing; see also the numbering of references because the guidelines indicate that should be in [].
Method: only move this section “2.1. Assessment of brain fog symptoms and quality of life” after the Participants and before procedures.
Move this section “3.1. Study population” to the sample information in Method, not in Results.
I suggest that authors write the hypotheses of the study in the method and, then, confirm/reject them (separately) in Discussion.
Many thanks and great work.
Author Response

(The authors gave the same response as above.)

Reviewer 4 Report
This article is good for publication in the journal; however, some of the points need to be addressed before the publication of this article. All the points have been mentioned below:
1. There are few grammatical mistakes in the manuscript, so check the complete manuscript completely to avoid such mistakes.
2. Add a few lines about causative agent SARS-CoV-2 & Please provide some information on the SARS-CoV-2 and routes of transmission of SARS-CoV-2 in the manuscript's introduction.
3. In the introduction section, read and cite - Protective immunity against COVID-19: Unravelling the evidences for humoral vs. cellular components. Travel Med Infect Dis 2021;39:101911.
4. The references are not as per the format of the journal. Update accordingly.
Rest is ok.
Author Response

(The authors gave the same response as above.)

Round 2
Reviewer 1 Report
Thanks a lot for responding. I believe I am now at a publishable level after having my edit history erased.